# Assessing the Relationship between Physical Activity and Depression in Lawyers and Law Professionals: A Cross-Sectional Study

Chapman Cox , Matthew S. Thiese * and Joseph A. Allen

Rocky Mountain Center for Occupational & Environmental Health, University of Utah Health, Salt Lake City, UT 84111, USA; chapman.cox@utah.edu (C.C.); joseph.a.allen@utah.edu (J.A.A.)
* Correspondence: matt.thiese@hsc.utah.edu

**Abstract:** Background: Law professionals are understudied, and little is known about their mental health. This cross-sectional study aimed to assess the relationship between the amount of physical activity and depressive symptoms. Methods: A sample of 681 law professionals completed a survey that assessed mental health conditions and activities to promote well-being. Crude and adjusted odds ratios (ORs) and 95% confidence interval (95% CI) were calculated for the relationship between the number of days of an hour or more of physical activity and two levels of depression from the patient health questionnaire-9 (PHQ-9). Results: Law professionals were found to have significantly more depressive symptoms, mild or worse and moderate or worse, when reporting 0 days of physical activity when compared to 7 days, with OR of 6.07 (95% CI 2.55–14.48) and 8.64 (95% CI 1.97–37.82) and adjusted ORs of 3.91 (95% CI 1.58–9.68) and 6.32 (95% CI 1.4–28.33), respectively. A dose response was also noted. Conclusions: There was a statistically significant relationship found between amounts of physical activity and depressive symptoms in law professionals. We recommend future research be conducted to better understand this relationship.

**Keywords:** law professional; lawyer; depression; physical activity; burnout; occupational health





## 1. Introduction

Lawyers play a fundamental role in most aspects of our society in politics, business, and the justice system that remains relatively understudied. Practicing law is an intellectually demanding profession, which requires competitive demands in the workplace such as high workloads and long working hours [1–3]. The few studies that have been performed have suggested that lawyers have problematic rates of alcohol use, depression, anxiety, stress, and suicidal ideation [2]. However, relatively little research has been published to better understand this occupational group's relationship with mental health concerns. Considering lawyers' impact on multiple aspects of our society, the potential relationships between their job and their health should be better understood.

The recent literature has shed a light on poor well-being outcomes in this population. Krill et al. found that among approximately 13,000 lawyers, depression prevalence was found to be 28% [2]. Prior reports from these data found the prevalence of major depressive disorder (MDD) to be 18% in law professionals, which included both lawyers and support staff, via the patient health questionnaire 9 (PHQ9) [4]. This is a high rate when compared to the general working population, which was found to show 5.1% [4]. This study also dove more deeply into the relationship between lawyers and support staff with depression. Lawyers by themselves (N = 554) were statistically significantly more likely to report mild, moderate, moderately severe, and severe depressive symptoms compared to the same working population (mild depressive symptoms OR = 3.89, moderate depressive symptoms OR = 5.29, moderately severe depressive symptoms OR = 9.71, and severe depressive symptoms OR 18.34, respectively).

Although there is a growing body of recent research focusing on law professionals, most of the research is focused on defining and better understanding where the negative outcomes lie and the nature of their severity as found in the aforementioned examples. Research on potential interventions or solutions to some of the problems found in law professionals focuses on topics including mindfulness and meditation [5–12]. The research does show efficacy for these modalities, but these methods are not accepted in the current guidelines for working with mental health outcomes [13,14], thus leaving a knowledge gap in understanding potential solutions or beneficial changes that can be made within this population that not only benefit the mental well-being but also the physical health of these individuals.

Psychological constructs may help explain the negative effect that appears to exist in lawyers' work-related demands. This manuscript will assess potential relationships between physical activity and depression through the lens of the Job Demands-Resource (JD-R) model [15]. In this model, physical activity would act as a resource that moderates the relationship between the demands placed on the worker from their job and the resulting occupationally related stress outcomes [16–18].

Furthermore, legal professionals' amount of physical activity may be a resource with a potential relationship with the onset of depression. In other words, physical activity may reduce the likelihood of experiencing the symptoms of depression. Multiple studies among other populations have found that there is an inverse relationship between physical activity and depressive symptoms, both for the treatment of depression and for prevention [19–26]. While this relationship has been established in other populations, this relationship has never been assessed among lawyers. Other observational studies suggest that physical activity (PA) is effective in reducing rates of depression and occupationally related psychological stress [27–30]. For example, a randomized control trial found significantly increased depression findings during periods of decreased activity compared to regular physical activity periods (FTime = 7.86, $p = 0.001$, $\eta^2 p = 0.19$) [31]. We believe that it is worth investigating how physical activity affects lawyers' well-being as it pertains to mental health.

We are also interested in understanding the potential relationship between suicidal ideation and physical activity. Thiese, Allen, Knudson, Free, and Petersen [4] found that law professionals were statistically significantly more likely to report suicidal ideation when compared to a general working population with an OR of 5.50 (95% CI 2.23 to 13.53). Due to the indicative nature of suicidal ideation and depression, it is an interesting relationship to explore.

This study aims to quantify the relationship, if one exists, between physical activity and depression found in law professionals. By gaining a better understanding of potential relationships between physical activity and depression prevalence in lawyers, these results may suggest future studies and programs to improve the mental health of these professionals. Given these arguments, we hypothesize that physical activity levels will be inversely related to both mild to severe and moderate to severe depressive symptoms in law professionals.

*Theoretical Background*

A theoretical framework previously mentioned, the Job Demands-Resources model, can provide a rationale for the relationship this study aims to assess. This framework, defined by Demerouti and Bakker [15], can be broken down into two elements. First, there are job demands, which are certain aspects of a job that require physical or psychological exertion, which in turn can lead to physiological and psychological costs [32]. Secondly, there are job resources that are aspects of the job that can decrease the costs related to the physical and psychological efforts of the job demands [15]. Also, they can be aspects that result in personal development and growth or aid in work goals. These can fall into several categories, including physical, psychological, social, or organizational.

The main hypothesis of this model states that if there are high levels of job demands and low levels of job resources, then workers can develop job strain. This hypothesis has

been supported by research findings that show higher levels of job demands can result in negative health outcomes in employees across different organizations [33,34]. Further research has even shown that the personal resource of compassion satisfaction can have moderative effects on the relationship between job demands and job strain [18].

Focusing on lawyers once more, the previous literature has identified serious demands in the work-lives of law professionals that negatively impact their mental health. For example, Wallace [35] demonstrated that time pressure to complete tasks is one of the highest contributors to negative mental health conditions in lawyers. Further, time billing, or the act of counting minutes towards specific work or projects, has been found to be another significant demand that has led to possible issues with mental health and substance abuse [1].

In terms of the resources that lawyers can experience throughout the duration of their career, there is a very common list of resources that have been investigated. This includes control over the amount of work, pay satisfaction, praise, promotional opportunities, job security, and the social value of work [1]. While these are important to take into consideration, it is equally valuable to consider resources that the lawyers have control over. These would be considered personal resources in the JD-R model. For example, the amount of time that lawyers typically allow towards self-care, such as therapy or physical activity, is seemingly low because of the high workload and low control over their own schedule. The previous literature has shown that personal resources mediate the relationship between job demands and the onset of psychological stress among workers [16,18,32,36].

## 2. Materials and Methods

These data were collected from a large study focusing on depressive symptoms and suicidal ideation in lawyers and other law professionals and were approved by the University of Utah Institutional Review Board. To take part in the study, participants gave online informed consent.

### 2.1. Data Collection

Methods for data collection have been described in greater detail in the study published previously [4,37]. Data collected on lawyers and legal support staff were used for this analysis.

The recruitment was performed by three different methods to obtain a large sample and minimize bias. This consisted of a randomized arm with a 53.9% response rate, advertisements through the Bar journal and publications, and inviting whole firms to participate. After collecting data from these three methods, we tested for equivalency amongst the different recruitment groups before combining data. Exclusion criteria included missing a response to a depression screener or the physical activity question (n = 13).

### 2.2. Questionnaire

Participants were electronically surveyed, after consenting, with a questionnaire through the REDCap software (ver. 14.0.31), by mail service, or over the phone. This survey consisted of 59 questions to better assess mental health outcomes and the utilization of available resources in this population. Measures that are of interest for this study include the patient health questionnaire 9 (PHQ-9), a validated depressive symptom form, and questions pertaining to recent physical activity over the previous week. Appendix A (in Table A1) and Appendix B include these questions from the questionnaire.

### 2.3. Physical Activity

Physical activity was assessed by asking a question "During the past 7 days, on how many days were you physically active for a total of at least 60 min per day? Add up all the time you spent in any kind of physical activity that increased your heart rate and made you breathe hard some of the time." and participants selecting a number of days. This was adopted by the NHANES questionnaire for physical activity and physical activity,

PAQ-706 [38]. Based upon the participant's response, they were placed in categories 0–7. The reference category was set for the highest amount of physical activity for ease of interpretation and because that is the most recommended level of activity.

### 2.4. Depression

The PHQ-9 is a validated tool used for screening the diagnosis of MDD needing clinical interventions [39–41]. It is a 9-question survey with four levels of response, resulting in possible scores ranging from 0 to 27. This battery has previously defined cutoff points for assessing no, minimal, mild, moderate, and severe depression, with a score of >10 indicating findings consistent with a diagnosis of MDD [40]. Previous research has demonstrated that the PHQ-9 is accurate for a clinical diagnosis meeting DSM-IV criteria for MDD with a specificity and sensitivity of 88% [40].

We dichotomized responses for the PHQ-9 into two depression outcomes. The first outcome classification that was dichotomized was to assess findings consistent with no or minimal depression versus mild to severe depression. The second outcome classification assessed a diagnosis of MDD based on the validated cutoff point in line with a clinical diagnosis of MDD, which resulted in the groups being no, minimal, or mild depression versus moderate to severe depression. We intended to use dichotomized variables for clinical importance, ease of interpretation, and to account for these data's non-linear nature.

### 2.5. Potential Confounders

An a priori list of confounders was decided based on previous publications. Known confounders of interest in this study are age, sex, and BMI, which were included in the adjustment model. Other potential confounders include drug and alcohol abuse, hours worked per week, type of law, and years in practice. We believe that these could potentially influence either the amount of exercise or the prevalence of depression. The known and potential confounders were assessed for a change in the relationship between physical activity and depression. The threshold was placed at 10% difference in the crude and adjusted odds ratios for these primary relationships.

### 2.6. Statistical Analysis

All data were analyzed using SAS 9.4 (SAS Institute, Cary, NC, USA). This is a statistical analysis tool that was created in the 1960s and has been developed into a software that is capable of handling robust statistical testing. Due to this software being well-trusted, commonly used, and capable of performing these tests, we selected it for our study.

To start, the mean and standard deviation were calculated for discrete and continuous variables. Frequency and percent were calculated for categorical variables. Three pairwise TOST tests assuming unequal variances (Satterthwaite) were used to test for equivalency in the variables of age, body mass index (BMI), and PHQ-9 across the different recruitment groups: those who were randomly selected, those who were self-selected, and participants from firms.

Simple logistic regression was used to provide the crude odds ratio (OR) and 95% confidence interval (CI), testing an association between physical activity and the two measures of depression prevalence. This regression compared the 8 levels of physical activity (0–7 days) versus the dichotomized depression variables. Multivariate logistic regression calculated adjusted odds ratios and 95% confidence intervals for the relationship between PA and depression prevalence, after controlling for confounders. We ran the adjusted model for the confounding variables of age, BMI, and gender. The potential confounders of drug and alcohol abuse, hours worked per week, type of law, and years in practice were assessed for a 10% or greater change in crude and adjusted odds ratios.

Using the discrete data from the PHQ-9, a linear relationship was assessed between physical activity and PHQ-9 depression scores using a generalized linear model. Post hoc analysis was conducted with a dichotomized variable of suicidal ideation for no vs. any presence of suicidal ideation in the previous 3 weeks. A regression was performed with

physical activity to understand if there was a potential relationship. Statistical significance was determined with a *p*-value of 0.05.

## 3. Results

Six hundred ninety-four participants completed and returned the questionnaire. Thirteen participants were excluded for missing responses. Table 1 displays the descriptive statistics for the sample and means, standard deviations, or frequency percentages for pertinent variables collected in this study. Age, BMI, and PHQ-9 were statistically equivalent for nearly all pair-wise comparisons between the three recruitment groups.

**Table 1.** Descriptive data for all law professionals.

| Demographics | Law Professionals N = 681 (Mean and SD or N and %) |
|---|---|
| Age | 47.61 (12.53) |
| BMI | 27.55 (6.01) |
| Gender | |
| Male | 387 (56.74%) |
| Female | 295 (43.26%) |
| Location | |
| Urban | 463 (67.99%) |
| Sub-Urban | 155 (22.76%) |
| Rural | 63 (9.25%) |
| Types of Law | |
| College or Law School | 4 (0.68%) |
| In house Attorney: Corporation of a for-profit institution | 42 (7.16%) |
| In house Attorney: Government, Public interest, or non-profit | 90 (15.33%) |
| Other law practice setting | 14 (2.39%) |
| Other setting (Not law practice) | 16 (2.73%) |
| Private Firm (All) | 348 (59.23%) |
| Private Firm (2–6 lawyers) | 68 (16.11%) |
| Private Firm (7–15 lawyers) | 43 (10.19%) |
| Private Firm (16–50 Lawyers) | 24 (5.69%) |
| Private Firm (51–99) | 20 (4.74%) |
| Private Firm (100–299) | 14 (3.32%) |
| Private Firm (300–749) | 14 (3.32%) |
| Sole practitioner private practice | 73 (12.44%) |
| Average Workweek | |
| Less than 30 | 95 (13.93%) |
| 31–40 | 169 (24.78%) |
| 41–50 | 287 (42.08%) |
| 51–60 | 103 (15.1%) |
| 61–80 | 25 (3.67%) |
| 81–100+ | 3 (0.34%) |
| Years Practicing | 18.15 (12.74) |
| Days Physically Active | |
| 0 days | 113 (16.57%) |
| 1 day | 105 (15.4%) |
| 2 days | 118 (17.3%) |
| 3 days | 113 (16.57%) |
| 4 days | 67 (9.82%) |
| 5 days | 74 (10.85%) |
| 6 days | 57 (8.36%) |
| 7 days | 35 (5.13%) |

**Table 1.** *Cont.*

| Demographics | Law Professionals N = 681 (Mean and SD or N and %) |
|---|---|
| PHQ9 | 5.34 (5.13) |
| PHQ9 Composite Score Categories | |
|    Minimal depression | 377 (55.28%) |
|    Mild | 181 (26.54%) |
|    Moderate | 70 (10.26%) |
|    Moderately Severe | 41 (6.01%) |
|    Severe | 13 (1.91%) |

The logistic regression analysis between physical activity and the two depressive outcomes' classification produced statistically significant results. Table 2 displays the crude odds ratio and 95% CI results of the logistic regression. The crude ORs comparing mild to severe depression with PA show significant association, with a statistically significant inverse monotonic dose–response relationship between physical activity and depressive symptoms. Similarly, the crude odds ratios between physical activity and moderate to severe depression suggest more exercise is associated with lower amounts of moderate to severe depression in line with a clinical diagnosis of MDD, with a statistically significant inverse monotonic dose–response. Physical activity had a statistically significant, inverse relationship with both defined outcome measures of depression and demonstrated a large dose response between these two relationships.

**Table 2.** Crude odds ratios and 95% confidence intervals for relationships between days of physical activity and either mild or worse depression or moderate or worse depression.

| Physical Activity | Mild to Severe Depression | | | Moderate to Severe Depression | | |
|---|---|---|---|---|---|---|
| | Odds Ratio | 95% Confidence Limit | | Odds Ratio | 95% Confidence Limit | |
| 0 Days | 6.07 [†] | 2.55 | 14.48 | 8.64 [†] | 1.97 | 37.82 |
| 1 Days | 4.55 [†] | 1.90 | 10.88 | 4.58 [†] | 1.02 | 20.55 |
| 2 Days | 4.08 [†] | 1.72 | 9.65 | 3.97 | 0.89 | 17.76 |
| 3 Days | 1.99 | 0.83 | 4.75 | 3.32 | 0.73 | 15.03 |
| 4 Days | 2.70 [†] | 1.08 | 6.75 | 4.14 | 0.88 | 19.45 |
| 5 Days | 1.74 | 0.69 | 4.37 | 1.27 | 0.23 | 6.87 |
| 6 Days | 1.18 | 0.44 | 3.17 | 1.68 | 0.31 | 9.16 |
| 7 Days | 1.00 | Reference | | 1.00 | Reference | |

[†] $p = 0.05$.

For confounding, only two of our suspected confounders were found to have a statistically significant association between both analyses of depression and physical activity. This was observed in mild to severe depression vs. PA for each year of increasing age (OR = 0.97, 95% CI 0.92–0.98) and each increase in kg/m$^2$ for BMI (OR = 1.06, 95% CI 1.03–1.09). There was no significant relationship for gender. In the adjusted analysis for moderate to severe depression vs. PA, the only significant confounder found was each increasing year of age (OR = 0.97, 95% CI 0.95–0.99). None of the other potential confounders, including drug and alcohol abuse, hours per week worked, types of law, and years in practice, were found to have a statistically significant relationship between our exposure and outcomes of interest. Thus, these were not included in the final adjusted model. After adjusting for our confounding variables of age, gender, and BMI, we observed that these play a significant role in the relationship between physical activity and depression. Table 3 shows the results for the multivariate regression that adjusts for our known con-

founding variables. There was still a statistically significant relationship found when accounting for confounding variables of age, BMI, and gender, with a statistically significant inverse monotonic dose–response relationship between physical activity and depressive symptoms but with meaningfully lower ORs between crude and adjusted analyses (e.g., 0 vs. 7 days crude OR = 6.07 vs. adjusted OR = 3.91). Less statistically significant relationships were found when adjusting for age, BMI, and gender with the cutoff points for MDD, 0 vs. 7 days (OR = 5.20, 95% CI 1.31–20.64). The statistical significance and dose response observed here strengthen the findings of an association between days physically active and reported symptoms of depression.

**Table 3.** Adjusted * odds ratios and 95% confidence intervals for relationships between days of physical activity and either mild or worse depression or moderate or worse depression.

| Physical Activity | Mild to Severe Depression | | | Moderate to Severe Depression | | |
|---|---|---|---|---|---|---|
| | Odds Ratio | 95% Confidence Limit | | Odds Ratio | 95% Confidence Limit | |
| 0 Days | 3.91 [†] | 1.58 | 9.68 | 6.32 [†] | 1.41 | 28.33 |
| 1 Days | 3.27 [†] | 1.33 | 8.06 | 3.66 | 0.81 | 16.59 |
| 2 Days | 2.81 [†] | 1.58 | 6.88 | 2.88 | 0.63 | 13.13 |
| 3 Days | 1.54 | 0.62 | 3.78 | 2.51 | 0.54 | 11.60 |
| 4 Days | 2.19 | 0.84 | 5.67 | 2.98 | 0.62 | 14.48 |
| 5 Days | 1.73 | 0.67 | 4.47 | 1.21 | 0.22 | 6.64 |
| 6 Days | 1.18 | 0.43 | 3.25 | 1.63 | 0.30 | 8.98 |
| 7 Days | 1.00 | Reference | | 1.00 | Reference | |

[†] $p = 0.05$ * Adjusted for age, gender, BMI.

Finally, we utilized the generalized linear model to test the discrete nature of the data from the PHQ-9 and PA levels ($p < 0.0001$). Through this analysis, further evidence was shown to support the hypothesis that, as the amount of weekly physical activity engaged in increases, the severity of depressive symptoms decreases. The regression conducted between physical activity and suicidal ideation resulted in minimal statistically significant findings. With a reference set at 6 days of physical activity, the crude odds ratio between no physical activity and any suicidal ideation was 5.34 (95% CI 1.37–20.79). Similarly, after adjusting for our known confounders of age, gender, and BMI, the adjusted odds ratio for this relationship was 5.43 (95% CI 1.37–21.43). These were the only statistically significant findings, and there was no observed dose response in this analysis.

## 4. Discussion

The aim of this study was to give more insight into the relationship between depression and the personal resource of physical activity found in the population of lawyers and law professionals. Both crude and adjusted odds ratios showed a dose response between decreased depressive symptoms and increased days physically active. Our data suggested that those who engage in less physical activity are more likely to experience depressive symptoms when compared to those who are physically active every day of the week (adjusted odds ratio; mild to severe depression of 0 vs. 7 days physically active OR = 3.91, 95% CI 1.58–9.68). The reference of 7 days was used because previous research has indicated the most physically active individuals were found to report the lowest levels of occupational-stress-related issues like burnout, anxiety, and depression [30].

These data suggest that there is a statistically significant inverse relationship among lawyers between physical activity and depressive symptoms and have shown there is legitimacy to physical activity and depressive symptoms as measured using a validated tool. This inverse relationship can also be expressed as follows: increased amounts of physical activity are associated with less depressive symptoms in this sample of lawyers

and law support staff. Therefore, we believe that lawyers may benefit from spending more time being physically active, and the demonstrated relationship should be promoted throughout this profession.

An interesting note is the impact of known confounders on this relationship. Although there are diminished odds ratio values in the adjusted analysis when compared to the crude odds ratios, there is still an observed dose response in this relationship.

There is significance shown across both pre-defined dichotomized variables of depression in our analyses. We believe that due to the fewer participants found in the MDD classification, overall there is less statistical power, which results in differences in monotonic dose response observed between the two depressive outcomes. However, there is still a trend for dose response in the relationships between physical activity and both classifications of depression, which is evidence to suggest that there is a significant relationship. The significance observed here is similar to findings between physical activity and depression in the previous literature [31,42]. This is the first study to look at this relationship in legal professionals and is not intended to be compared to other occupational groups. We believe this is worth considering for future research and recommending for this population to potentially observe lowered prevalence of depression.

The dose response observed in both dichotomized variables of depression is backed by the tests conducted in the secondary analysis utilizing the generalized linear model. The results for this analysis are not shown due to their exploratory nature. By using this test, we are able to observe the PHQ-9 data in the discrete manner they were collected in and not limited to a dichotomized fashion. However, this test did show a strong predictive value for the relationship between the PHQ-9 scores and the levels of physical activity by day. With statistical significance, this model demonstrated that, as the number of days physically active increased in participants, the predicted score of depressive symptoms decreased with low variance. By performing this test, our aim was to take a broader look at the data we had provided to see if our claims were valid. And as a result, this test showed that the previous tests are corroborated, and this improves the argument for the importance of physical activity in this group of law professionals.

Further, when taking the previous literature into account, there is a stronger case made for the importance of physical activity within this population. For example, a study looking at the relationship between cardiorespiratory fitness and burnout and depression symptoms found a similar dose–response relationship as our study [28]. These findings support the validity of our findings and highlight the importance of considering the importance of the role physical activity plays in the poor mental health outcomes found in this population. This, in turn, emphasizes the need for further research to be conducted to better understand just how impactful this resource could be with potential temporal style investigations.

### 4.1. Implications for Theory

Our study has implications for theory. The first major implication pertains to how this study provides evidence in support of the JD-R model [15]. This model suggests that as job demands increase (e.g., high workload), the amount of perceived worker stress increases as well, potentially leading to depression, anxiety, and substance abuse. The JD-R model postulates that workers have resources at their disposal to respond to the demands. These are aspects of their job or life that help provide relief or compensation for the work, including the stress that may arise from excessive demands [15]. One example of a resource is the control that workers have over their work [1]. This control or autonomy is a resource that reduces stress among those who have this resource. However, many workers have little control over their work environment and, as that control decreases, the amount of lingering stress increases.

Due to the low level of resources and high demand in the law profession [1,2,35], it is intuitive to aim to give more control back to lawyers over their self-care and mental health through physical activity. There is evidence to suggest that physical activity can buffer the effects caused by high job demands in a broad, non-occupation-specific manner [43]. We

believe that the results of this analysis demonstrate the ability for physical activity to act as a personal resource in the occupational group of law professionals based on the previously discussed dose response. Further, because this is a potential resource that is engaged in typically in the personal time of the employee, this aids in the argument that personal resources are effective in moderating the relationship between demands and occupationally related stress.

A second theoretical implication for why the tested relationships exist may include biological mechanisms. By increasing days physically active, individuals experience an increase in the release of endorphins which are known to improve mood and mental health [44]. Regular physical activity or exercise has been shown to be protective against oxidative stress, and oxidative stress may be related to the pathophysiology of depression [45,46]. While these are not theories we were testing, the biological plausibility aids in our claims for an inverse relationship between physical activity and depression.

### 4.2. Practical Implications

Given the findings in this analysis, lawyers should be encouraged to engage in physical activity more frequently. This is especially true for those who have a sedentary nature of work, as there is no additional activity being introduced through their daily work routine.

Possible interventions could include employee education programs in the workplace that are shared through newsletters, classes, or meetings. The CDC recommends including various forms of social support programs to improve employee involvement in increasing activity [47].

This study can support the claim for the practical application of encouraging physical activity in lawyers and other law professionals. Because of the nature of physical activity being a personal resource, we believe that those in the occupational field of law should consider PA and other personal resources to improve the symptoms of job-demand-related stress.

### 4.3. Limitations and Future Directions

For limitations, there are a few due to the cross-sectional study design. First, more testing should be conducted in other regions or states to verify these findings. Also, there were fewer data on the support staff compared to lawyers, so enrolling more support staff would improve the generalizability to this population as well. Data were collected at one time point, so we are unable to make causal inferences in the tested relationship. There is also potential for a healthy worker effect in this sample.

For data collection, we were limited to the measures utilized in the original questionnaire. The PHQ-9 works well as a validated tool, but we were limited to a continuous scale question of physical activity. Ideally, we would be able to monitor activity in participants, but practically, it would be better to use a previously validated tool. Finally, we were able to control for known confounders in our multivariate regression and test for potential confounders. One confounder we were not able to test for was disability status, which could meaningfully impact this relationship.

Future research should be conducted to see the temporal affect physical activity has on the outcome of depression and other mental health issues found in this population with the use of intervention studies. Also, other potential personal resources should be hypothesized and studied for their effect on mental health outcomes in law professionals.

### 5. Conclusions

This analysis demonstrates physical activity having a meaningful relationship with depression found in lawyers and other law professionals, as this is a prevalent issue in this occupational group [2,4]. It also explores the concept of personal resources and the effect they have on outcomes stemming from high job demands found in law. The findings from the crude and adjusted analyses in this study have shown a significant dose–response relationship between the amount of weekly physical activity and fewer reports of depressive

symptoms found in law professionals. In addition, the relationship is emphasized due to the strongest association being found between those who classify as having higher levels of depression and no exercise. Our evidence suggests that more investigation should be conducted looking at the impact of physical activity on this population, as well as other personal resources with potential relationships to symptoms of stressors from the intense job demands found in the field of law. The data presented in this study help further the growing body of literature focusing on the poor outcomes found in law professionals. Future research should be conducted to assess the effectiveness of implementing physical activity programs for law professionals, but these findings suggest efficacy in encouraging physical activity for those in the legal field.

**Author Contributions:** Conceptualization, C.C., M.S.T. and J.A.A.; methodology, C.C. and M.S.T.; validation, C.C., M.S.T. and J.A.A.; formal analysis, C.C. and M.S.T.; investigation, C.C., M.S.T. and J.A.A.; resources, M.S.T. and J.A.A.; data curation, M.S.T.; writing—original draft preparation, C.C.; writing—review and editing, C.C., M.S.T. and J.A.A.; visualization, C.C.; supervision, M.S.T. and J.A.A.; project administration, C.C.; funding acquisition, M.S.T. All authors have read and agreed to the published version of the manuscript.

**Funding:** This research was supported by a grant from the Centers for Disease Control and Prevention (NIOSH) NIOSH Education and Research Center training grant T42/CCT810426-10, National Center for Advancing Translational Sciences (NCATS/NIH) 8UL1TR000105, and the Utah State Bar.

**Institutional Review Board Statement:** The parent study from which this data was utilized from was approved by the University of Utah Institutional Review Board (Approval Code: 120539, and Approval Date: 22 April 2019).

**Informed Consent Statement:** To take part in the study, participants gave online informed consent.

**Data Availability Statement:** We are unable to share data freely because it contains Personal Identifiable Information. For further inquiries, please contact the corresponding author.

**Acknowledgments:** We would like to thank the Utah State Bar for helping produce this work.

**Conflicts of Interest:** The authors declare no conflicts of interest.

## Appendix A  Patient Health Questionnaire 9

**Table A1.** Over the last 2 weeks, how often have you experienced any of the following problems?

|  |  | Not At All | Several Days | More than Half the Days | Nearly Every Day |
|---|---|---|---|---|---|
| 1. | Little interest or pleasure in doing things |  |  |  |  |
| 2. | Feeling down, depressed, or hopeless |  |  |  |  |
| 3. | Trouble falling or staying asleep, or sleeping too much |  |  |  |  |
| 4. | Feeling tired or having little energy |  |  |  |  |
| 5. | Poor appetite or overeating |  |  |  |  |
| 6. | Feeling bad about yourself—or that you are a failure or have let yourself or your family down |  |  |  |  |

**Table A1.** *Cont.*

| | | Not At All | Several Days | More than Half the Days | Nearly Every Day |
|---|---|---|---|---|---|
| 7. | Trouble concentrating on things, such as reading the newspaper or watching television | | | | |
| 8. | Moving or speaking so slowly that other people could have noticed? Or the opposite—being so fidgety or restless that you have been moving around a lot more than usual | | | | |
| 9. | Thoughts that you would be better off dead or of hurting yourself in some way | | | | |
| 10. | Feeling Nervous, Anxious, or on edge | | | | |

**Appendix B**

During the last 7 days, on how many days did you do moderate to vigorous physical activities/exercise for 20 min or more?

☐ 0 Days
☐ 1 Day
☐ 2 Days
☐ 3 Days
☐ 4 Days
☐ 5 Days
☐ 6 Days
☐ 7 Days

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
