# Peer review of "Assessing the Relationship between Physical Activity and Depression in Lawyers and Law Professionals: A Cross-Sectional Study"

_merits, doi:10.3390/merits4030017_

Round 1

Reviewer 1 Report

Comments and Suggestions for Authors

1.The research topic is meaningful and helpful to lawyers.

2.What do the existing knowledge gape connecting with this study aims? Please introduce more specific statements and supportive references.

Author Response

Comment 1: The research topic is meaningful and helpful to lawyers.

Response 1: Thank you for this positive comment, we appreciate that!

Comment 2: What do the existing knowledge gape connecting with this study aims? Please introduce more specific statements and supportive references.

Response 2: This was a very insight comment, thank you for pointing this out. We have addressed this in the revised draft by adding to the introduction of this paper. These changes introducing specifically what gap we are addressing can be found in lines 43-52 of the new draft. 

Reviewer 2 Report

Comments and Suggestions for Authors

Thank you for the opportunity to review this interesting paper. In my opinion, the manuscript is well written, the objective is clear, and the selected method is appropriate for the purpose,  but there is room for improvement. Here you have my recommendations:

1) After the introduction, a specific section devoted to the theoretical background is required. Please explain the JD-R model, and convincingly defend your positioning.

2) Please, consider the inclusion of "your questionnaire" as an Annex of the work signaling the construct/questions related to PHQ-9, PAQ-706... That should be good for gaining soundness, for the replicability of your work by potential readers, and for gaining citations.

3) Please, include the meaning of BMI.

4) The reference to have been using SAS 9.4 must be at the beginning of the section, not at the end. In addition, please, consider offering a paragraph explaining what is SAS, why you chose it, and its benefits.

I hope my comments and recommendations help you to up grade your work.

Kind regards

Author Response

Comment 1: After the introduction, a specific section devoted to the theoretical background is required. Please explain the JD-R model, and convincingly defend your positioning.

Response 1: Thank you for this comment, we have taken this into consideration and addressed this in our revised draft. We have added a Theoretical Background section to the end of the introduction of the draft. This can be found in lines 85-117. 

Comment 2: Please, consider the inclusion of "your questionnaire" as an Annex of the work signaling the construct/questions related to PHQ-9, PAQ-706... That should be good for gaining soundness, for the replicability of your work by potential readers, and for gaining citations.

Response 2: We appreciate the insight! There has been copies of the questions added to the draft in Appendices A and B. 

Comment 3: Please, include the meaning of BMI.

Response 3: Thank you for pointing this out, we have fixed this in the revised draft on line 181. 

Comment 4: The reference to have been using SAS 9.4 must be at the beginning of the section, not at the end. In addition, please, consider offering a paragraph explaining what is SAS, why you chose it, and its benefits.

Response 4: Agreed, we have switched this to be addressed at the beginning of the statistical analysis section in lines 175-178. 

Reviewer 3 Report

Comments and Suggestions for Authors

The topic discussed in this study is important and interesting.

The structure of the paper is adequate as well as the methods used.

My overall opinion about this study is positive - the paper is interesting, not exciting but interesting.

However, I have three main concerns:

- I feel the lack of some literature review because what is presented in the Introduction is not enough;

- Regarding the interpretation of the evidence, it could be stronger, namely through a closer link with previous studies on related topics;

- The conclusion is poor and needs a careful revision.

Author Response

Comment 1: I feel the lack of some literature review because what is presented in the Introduction is not enough

Response 1: Thank you for pointing this out, and we agree that this needs to be addressed. We have done so by adding some literature review/addressing what solutions are currently being sought in this group and theoretical background to the introduction section in lines 43-52 and 85-117. 

Comment 2: Regarding the interpretation of the evidence, it could be stronger, namely through a closer link with previous studies on related topics

Response 2: Thank you for this comment, I/we did not want to feel like we were making too strong of claims with this evidence but agree that this could be improved upon. Therefore, we have added a paragraph within the discussion (lines 317-325) to address this and point to research that had very similar findings. 

Comment 3: The conclusion is poor and needs a careful revision.

Response 3: We appreciate this comment, and hope that the revised version of the conclusion in the new draft is an improvement. Please let me/us know if there are any more specific things you would like to see addressed to improve the quality of this conclusion. 

Round 2

Reviewer 2 Report

Comments and Suggestions for Authors

Thank you for taking into consideration my recommendations. I think that the manuscript has improved and could be published. 

Reviewer 3 Report

Comments and Suggestions for Authors

The new version of the paper answers my previous concerns. Therefore, in my opinion, the paper can be accepted for publication.